# HFMRE: Constructing Huffman Tree in Bags to Find Excellent Instances for Distantly Supervised Relation Extraction

**Min Li**[2,1]*, **Cong Shao**[1]* , **Gang Li**[1,2]†, **Mingle Zhou**[1]

[1] Key Laboratory of Computing Power Network and Information Security, Ministry of Education, Shandong Computer Science Center (National Supercomputer Center in Jinan), Qilu University of Technology (Shandong Academy of Sciences), Jinan, China

[2] Faculty of Data Science, City University of Macau, Macau, China

`limin@qlu.edu.cn, 10431220590@stu.qlu.edu.cn`

`lig@qlu.edu.cn, zhouml@qlu.edu.cn`

## Abstract

Since the introduction of distantly supervised relation extraction methods, numerous approaches have been developed, the most representative of which is multi-instance learning (MIL). To find reliable features that are most representative of multi-instance bags, aggregation strategies such as AVG (average), ONE (at least one), and ATT (sentence-level attention) are commonly used. These strategies tend to train third-party vectors to select sentence-level features, leaving it to the third party to decide/identify what is noise, ignoring the intrinsic associations that naturally exist from sentence to sentence. In this paper, we propose the concept of circular cosine similarity, which is used to explicitly show the intrinsic associations between sentences within a bag. We also consider the previous methods to be a crude denoising process as they are interrupted and do not have a continuous noise detection procedure. Following this consideration, we implement a relation extraction framework (HFMRE) that relies on the Huffman tree, where sentences are considered as leaf nodes and circular cosine similarity are considered as node weights. HFMRE can continuously and iteratively discriminate noise and aggregated features during the construction of the Huffman tree, eventually finding an excellent instance that is representative of a bag-level feature. The experiments demonstrate the remarkable effectiveness of our method[1], outperforming previously advanced baselines on the popular DSRE datasets.

## 1 Introduction

In natural language understanding tasks, relation extraction is an important fundamental task. Its goal is to understand the contextual information

---

\* Equal Contribution

† Corresponding author

[1]The source code and data can be available at https://github.com/shaocong-qy/HFMRE

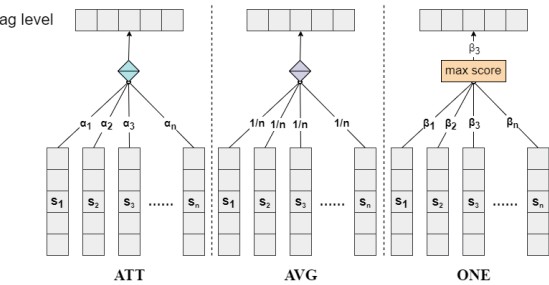

Figure 1: Three aggregation strategies in the MIL framework.

of a sentence or paragraph to identify the target relationships of entities in the text. Translating people's perceptions of the physical world into semantic information that a computer can understand in a structured format. Traditional relation extraction methods currently rely on large-scale, high-quality datasets, which are often labor-intensive and time-consuming. To address the difficulty of obtaining high-quality annotated samples, Mintz et al. (2009) et al. proposed a distantly supervised relation extraction algorithm. They make an important assumption: for a triad in an existing knowledge graph, it is assumed that any sentence in an external document repository (e.g. Wikipedia) that contains the pair of entities reflects this relation to some extent. Based on this assumption, the distantly supervised algorithm can label sentences in an external document repository with relations based on an annotated mini-knowledge graph, which is equivalent to doing automatic annotation of samples. However, this assumption suffers from data annotation problems, since a sentence may contain only this entity pair, but does not express the relationship in Freebase.

To alleviate the noise problem present in the dataset, Zeng et al. (2015) et al. first combined multi-instance learning with deep learning, which is now the dominant learning framework. Given a set of multi-instance bags with categorical labels,

each bag contains several sentences without categorical labels. The model builds a multi-instance classifier by learning from multi-instance bags and applies this classifier to the prediction of unknown multi-instance bags. There are three main methods used to reduce or eliminate noisy data from the bag during MIL: Average (AVG) (Gao et al., 2021), At-Last-One (ONE) (Zeng et al., 2015) and Sentence-Level Attention (ATT) (Lin et al., 2016). As shown in Figure 1, the AVG is an average representation of all sentences in the bag as a feature representation of the bag. The ONE is to first select the highest confidence sentence in each bag, and then use only the highest confidence sentence as a feature of the bag for subsequent training. The ATT uses a selective attention mechanism that enables the model to assign different weights to sentences in a bag, using a weighted average of the sentence-level representations to produce a bag-level representation.

All of the above methods have some weaknesses. The AVG uses an averaging operation that treats each sentence in the bag equally and inevitably introduces noise into the bag-level feature representation, causing a degradation of model performance. The ONE, based on the at-least-one assumption, selects the highest scoring sentence in the bag as the bag-level feature representation, avoiding the effect of noise as much as possible, but not making full use of the data in the bag, resulting in a waste of data resources. The ATT uses learnable attention to selectively assign weights to sentences, and although it compensates for the shortcomings of AVG and ONE, however, we believe that this is not an optimal aggregation strategy, as it does not take into account information about the correlation between sentences, and each sentence exists in isolation within the bag.

We hold one hypothesis — the inherent association between sentences within the same bag is more helpful in discriminating noise. Based on this opinion, a simple and effective baseline has been developed. First, the intrinsic association of the two sentences is visualized using cosine similarity. However, we are not the first to use cosine similarity to reduce the effect of noise, Yuan et al. (2019) et al. calculate the similarity between sentences and relation vectors to capture the correlation between them, and thus reduce the number of sentence instances that are not correlated with the relation facts. This is an indirect approach, where

the relation vectors are trained, and are likely to be influenced by other factors in the training process, and it is largely impossible to train a perfect relation vector. We use a more direct and effective way to identify the noise, called the circular cosine similarity method (see section 3.2). Simply put, it is a loop that calculates the sum of the similarity of each sentence in the bag to other sentences. The greater the sum of the similarity of a sentence, the more relevant it is to other sentences and the more likely it is to be a positive example, and vice versa. Then, to exploit this correlation information between sentences, we consider each sentence in the bag as a node needed to construct a Huffman tree based on the idea of Huffman trees, and use the circular cosine similarity as node weights to iteratively identify noise and aggregate sentence features by merging old nodes to generate new nodes, and finally construct an excellent instance that can represent the features of the bag.

As the paper uses the Huffman tree idea to solve the relation extraction problem, we name our baseline model HFMRE. By constructing Huffman trees to reduce noise, finding excellent instances within bags has reasonable interpretability: A Huffman tree is the tree with the shortest path with weights, the higher the weight, the closer to the root node. Extending to the Huffman tree in the paper, the circular cosine similarity is used as the weight of the nodes, so that the positive instances in the bag are close to the root node and the negative instances are far from it. When aggregating sentence features, the negative instances are then fused one step ahead of the positive instances. Since the aggregation of features is a continuous updating process, as shown in Figure 2, the positive instances express more features in the root node, and the noise impact of the negative instances is significantly reduced. To validate the effectiveness of HFMRE, we evaluated our model on three mainstream DSRE datasets, including NYT10d, NYT10m and wiki20m. The results showed that HFMRE significantly outperformed the current state-of-the-art baseline, increasing the AUC by 3.2% and boosting the P@M to 86.4. In addition, the ablation study showed the individual contribution of each module.

Overall, the main contributions of the paper are summarized as follows:

1. We propose a baseline model HFMRE for the DSRE task. It uses the idea of the Huffman

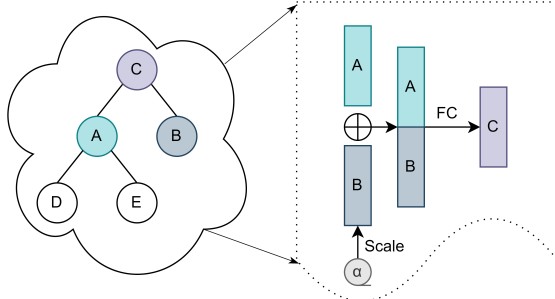

Figure 2: An example of nodes fusion during Huffman tree construction. The right child is the one with the lower weight by default.

tree algorithm, which treats sentences as nodes to be selected, discovers noise and aggregates sentence information in the process of constructing a Huffman tree, and ultimately constructs excellent instances that express bag-level features.

2. The concept of circular cosine similarity is proposed. Compared with the selective attention mechanism, circular cosine similarity can directly, quickly and effectively discriminate potentially noisy data in the bag, and thus effectively guide the model to selectively aggregate sentence features.

3. The experiments demonstrate that our method outperforms previous baselines, achieving state-of-the-art results on three datasets and providing a robust baseline model for the DSRE task.

## 2 Related Work

### 2.1 Distantly Supervised Relation Extraction

Distantly supervised relation extraction was proposed as a solution to the problem that large amounts of high-quality annotated data are difficult and costly to obtain. The main learning framework currently available is multi-instance learning, which was first applied to relation extraction by Hoffmann et al. (2011) et al. The method was soon used in most neural models because of its effectiveness in alleviating the noise problem in the dataset. E.g., neural networks based on CNN(Zeng et al., 2015; Jiang et al., 2016), Transformer(Alt et al., 2019), GNN(Vashishth et al., 2018), BERT(Christou and Tsoumakas, 2021), etc. Although MIL's pattern can greatly reduce DS noise, it does not represent many other useful sentence features in the dataset. Therefore many other frameworks have recently been derived based on MIL. (1) Contrastive learning framework. Chen

et al. (2021) et al. propose a contrastive instance learning framework that uses the initial MIL as an encoder of relational triples and constraint positive pairs against negative pairs for each instance. So as to improve the performance of DSRE models in the MIL framework. (2) Reinforcement learning framework. Qin et al. (2018) et al. propose a deep reinforcement learning framework that can be applied to any state-of-the-art relation extractor. Feng et al. (2018) et al. define instance selection as a reinforcement learning problem, where the instance selector selects high-quality sentences by reinforcement learning. The relation classifier makes predictions about the sentences, providing rewards to the instance selector. (3) Hierarchical Contrastive Learning Framework. Li et al. (2022) et al. propose a hierarchical contrastive learning framework for the DSRE task. It can make full use of semantic interactions within specific levels and cross levels to reduce the impact of noisy data.

### 2.2 Huffman Tree

The Huffman tree was first proposed by Huffman (1952), also known as an optimal binary tree, for a given N leaf node with weighted values, the aim is to construct a minimum binary tree with weighted path lengths. The Huffman tree is a data structure that is commonly used for data compression and coding. Later, Huffman trees were also used in other areas such as channel coding(Yin et al., 2021; Liu et al., 2018; Wu et al., 2012), text compression(Dath and Panicker, 2017; Bedruz and Quiros, 2015; Mantoro et al., 2017), image compression(Yuan and Hu, 2019; Kasban and Hashima, 2019; Patel et al., 2016), audio coding(Yi et al., 2019; Yan and Wang, 2011), etc.

In recent years, with the development of deep learning, the idea of Huffman trees has been introduced. Morin and Bengio (2005) et al. was first proposed using Huffman trees for Hierarchical Softmax, and the method was subsequently widely used and developed by later generations(Mnih and Hinton, 2008; Chen et al., 2016; Mikolov et al.). With the rise of large deep neural network models, Gajjala et al. (2020) et al. train large deep neural network (DNN) models in a distributed manner based on Huffman coding techniques. Due to the computationally intensive and memory-intensive nature of neural networks, which makes them difficult to deploy on embedded systems with limited hardware resources, Huffman coding was intro-

duced to enable model compression. Han et al. et al. apply a three-stage pipeline: pruning, trained quantization and Huffman coding. The network is first pruned by learning only the critical connections. Next, the weights are quantized to force weight sharing and finally, Huffman coding is used to compress the parameter files. Wang et al. (2020) et al. proposed a compression schema (HEPC) based on Huffman coding, which can reduce the kernel schema number and index storage overhead of sparse neural networks with little loss of accuracy. Today, Huffman trees have been used in a variety of fields, for example, in natural language processing, Khan et al. (2020) et al. to develop fixed output length text codes using Huffman's algorithm and to construct new vocabularies based on codewords. In the field of biology, Jin et al. (2016) et al. introduced Huffman coding to calculate the similarity of DNA sequences.

## 3 Methodology

We treat each sentence in the bag as a node to be selected that we need to construct a Huffman tree, so each bag eventually generates a Huffman tree and then looks through the Huffman tree to find excellent instances. The Huffman tree constructed in the method has the following characteristics:

*1) The parent nodes in a Huffman tree aggregate the features of the child nodes.*

*2) The root node of a Huffman tree aggregates the features of all sentences within the bag and is used as the bag feature for relation querying.*

*3) A sentence with more reflected bag features (excellent instances) is closer to the root node and expresses more features in the root node.*

The main architecture of the HFMRE model is shown in Figure 3. There are four major steps: sentence encoding, constructing Huffman tree nodes, constructing Huffman trees and relation querying. The specific details are described below.

### 3.1 Sentence encoding

Specifically, we employ a bag $B_i(e_1, e_2) = \{I_{i,1}, I_{i,2}, \cdots, I_{i,n}\}$ as a unit to encode sentence features, each containing $n$ sentences. For each sentence $I_{i,j} = \left(\mathbf{x}_1^j, \mathbf{x}_2^j, \cdots, \mathbf{x}_n^j\right)$ in the bag, we add the special token "[CLS]" at the beginning of the sentence, and the head and tail entities in the sentence are surrounded by the special symbols "$" 

and "#"(Wu and He, 2019). Thus, the sentence is processed before being fed to the encoder as:

$$I_{i,j} = ([\text{CLS}], \cdots, \$, \cdots, \$, \cdots, \#, \cdots, \#, \cdots) \tag{1}$$

We input the sentences to the encoder (BERT) to generate a context-rich semantic embedding $\hat{I}_{i,j}$ containing each token $x$, then followed the setting of Devlin et al. (2019) et al. using the final hidden state corresponding to the special token "[CLS]" as the aggregated sequence representation of the relation extraction task.

### 3.2 Constructing Huffman Tree Nodes

In this section, we will prepare the leaf nodes needed for the Huffman tree algorithm. It is well known that each node to be selected in a Huffman tree has an information frequency (often recorded as a weight) and a name that belongs to it. We therefore make three important assumptions to make the conversion of sentences to leaf nodes:

*a. Assume that the order of the sentence in the bag is the "name" attribute of the node.*

*b. Assume that the circular cosine similarity of the sentence is the "weight" attribute of the node.*

*c. Further, we add a new attribute "value" to the node, and assume that the encoded feature of the sentence is the "value" attribute of the node, which is used as a feature vector when aggregating excellent instances within the bag.*

In conclusion, this is the conceptual definition of the nodes of the Huffman tree in this study. The conceptual diagram of the nodes is shown in Figure 3 and the formula is expressed as follows:

$$N_i \langle name | weight | value \rangle \tag{2}$$

The circular cosine similarity is the sum of the cosine similarity of a sentence to all other sentences in the bag. As shown in Figure 3, each sentence in the bag is calculated against the other sentences, just like in a round-robin football game, where each team plays each other once. Ultimately, we take the sum of the similarity of a sentence to other sentences as the circular cosine similarity of that sentence. Let a bag $B_i(e_1, e_2) = \{I_{i,1}, I_{i,2}, \cdots, I_{i,n}\}$ contains n sentences, then the circular cosine similarity of the $j_{th}$ sentence is defined as:

$$\text{CCSC}_j = \sum_{x=1}^{n} \frac{\hat{I}_{i,j} * \hat{I}_{i,x}}{||\hat{I}_{i,j}|| ||\hat{I}_{i,x}||} \tag{3}$$

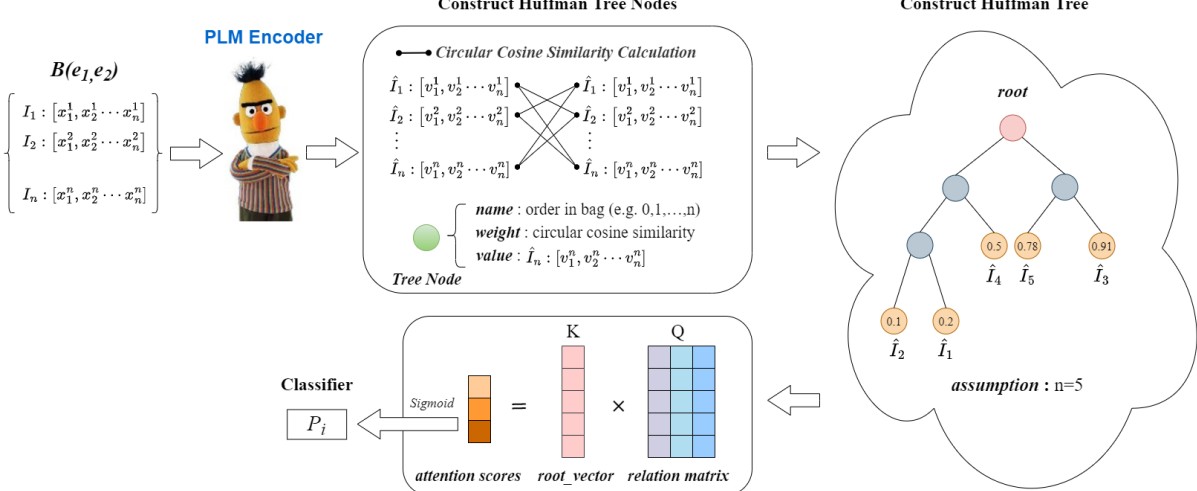

Figure 3: Model architecture for HFMRE.

There is some conceptual advantage in using circular cosine similarity as the weight of a Huffman tree node, which essentially reflects the overall relevance of a sentence to other sentences. Assuming that sentence *A* has a higher circular cosine similarity than sentence *B*, then sentence *A* is a better representation of the whole bag than sentence *B*, because sentence *A* is closer to the other sentences and jointly represents the bag features. Moreover, when a sentence has significantly lower circular similarity than other sentences, then there is a high probability that the sentence is a noisy piece of data.

### 3.3 Constructing a Huffman Tree

Huffman tree, originally a data structure for computer numerical storage, is used to compress and encode data. Its construction is a continuous iterative process, merging the two binary trees with the smallest weight each time, and the weight of the merged tree is the sum of the weight of the first two trees, so the structure of Huffman tree is a bit like a pyramid, and the node at the top has the largest weight.

In this paper, we follow the algorithmic idea of Huffman trees. The two sentences with the smallest weight (circular cosine similarity) are fused each time in the set of nodes $list\{node_1, node_2, \cdots, node_n\}$, where the sentence with the smaller weight is multiplied by a reduction factor $\alpha$, which is used to control the effect of noise. As shown in Figure 2. The new node after fusion can be formulated as:

$$N_{new}\langle name\rangle = \mathrm{arg}Max\{N_{1,2,...,n}\langle name\rangle\} + 1 \quad (4)$$

$$N_{new}\langle value\rangle = \sigma\big([N_i\langle value\rangle \oplus N_j\langle value\rangle * \alpha] * W_s\big) + b_s \quad (5)$$

$$N_{new}\langle weight\rangle = \sum_{x=1}^{n} \frac{N_{new}\langle value\rangle * N_x\langle value\rangle}{||N_{new}\langle value\rangle||||N_x\langle value\rangle||} \quad (6)$$

where $\sigma$ denotes the non-linear activation function, $\oplus$ denotes vector connection, $W_s$ denotes a trainable parameter matrix, and $b_s$ is the bias, both of which are shared globally, $\alpha$ is a hyperparameter, and n denotes the total number of nodes in the set.

It is worth noting that due to the merging of two nodes into a new node (subtree), the set of nodes is thus updated — old nodes are deleted and new nodes are added. In equation (6), the weights of the new node are the circular similarity between the sentences in the updated set of nodes, and likewise, the weights of the other nodes to be selected in the set are updated at the same time.

The above steps are repeated until there is only one node left in the set *list*, i.e. the root node of the Huffman tree, $root\langle name|weight|value\rangle$, and the Huffman tree construction is complete. Based on the three traits of the Huffman tree in our method, the root node continuously aggregates the features of the outstanding instances in the bag, so we use the feature $root\langle value\rangle$ of the root node for subsequent relation querying. The code implementation of this process can be found in the Appendix D.

### 3.4 Relation Querying

We trained a query vector $q_i$ for each relation $r_i$ separately, as shown in Figure 3, and all the query vectors form a relation matrix, denoted as $Q$. The root node constructed in the above process is then denoted as $K$. The attention score of each query vector $q_i$ and root$\langle value \rangle$ is calculated based on the idea of attention, and then activated by *sigmoid* function, the probability $p_i$ of root$\langle value \rangle$ and each relation $r_i$ is returned. In inference, a positive prediction is performed if $p_i >$ threshold (0.5). The process is denoted as:

$$P(K|Q) = sigmoid(QK^T) \tag{7}$$

### 3.5 Loss Function

We use a simple binary cross-entropy loss function without adding any loss term to evaluate the difference between the predicted label and the gold label for each bag.

## 4 Experiments and Analysis

We conduct experiments on popular datasets from the DSRE task to validate the effectiveness of our method. The experimental results were analyzed in detail and ablation experiments were performed to understand the key factors affecting the performance of the model.

### 4.1 Datasets

We evaluated our method HFMRE on three popular DSRE benchmarks — NYT10d, NYT10m and Wiki20m, and the statistics for the datasets are presented in Appendix C.

**NYT10d** is the most popular dataset used on distantly supervised relation extraction tasks from Riedel et al. (2010) et al. The dataset was distantly supervised by NYT corpus with Freebase.

**NYT10m** is a dataset recently published by Gao et al. (2021) et al. It merges some inappropriate relation categories from NYT10 to form a 25-category relation dataset and contains a manually annotated test set.

**Wiki20m** is also published by Gao et al. (2021) et al. Its test set was derived from the supervised dataset Wiki80, and the training/validation/test set was repartitioned to ensure that no identical sentence instances existed between the three partitions to avoid any information leakage.

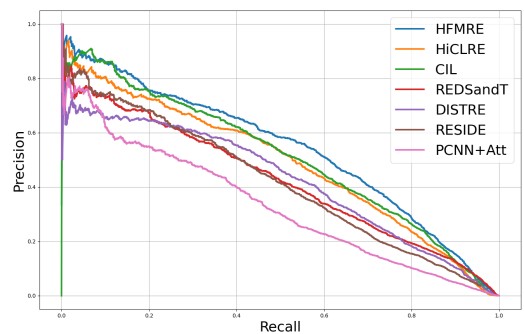

Figure 4: PR Curve for Models on NYT10d.

### 4.2 Evaluation Parameters and Hyper-parameters

We follow the previous setup and use evaluation parameters that are widely used in the literature. These include P@N (P@100, P@200, P@300): indicating the top N most confident predictive precisions computed by a model on a triad; P@M: indicating the average of the three P@N results above; AUC: indicating the area under the ROC curve, meaning the classifier's ability to classify positive and negative cases; M-F1: macro f1; $\mu$-F1: micro f1, and we also plotted PR curves to show the trade-off between model precision and recall.

We use bert-base-uncased as the pre-training weights for our model HFMRE, and the detailed settings of the hyper-parameters are given in Appendix A.

### 4.3 Baseline Models

For the NYT10d dataset, we compared some of the models that are representative of recent years. Including PCNN-ATT(Lin et al., 2016), RESIDE(Vashishth et al., 2018), DISTRE(Alt et al., 2019), REDSandT(Christou and Tsoumakas, 2021), CIL(Chen et al., 2021) and HiCLRE(Li et al., 2022). For the recently released datasets NYT10m and Wiki20m, we report the comparison results with the baseline models BERT-ONE, BERT-AVG, and BERT-ATT from the original paper(Gao et al., 2021), and we additionally add comparisons with the more robust model CIL and the latest model HICLE.

### 4.4 Evaluation on Distantly Supervised Set

Table 1 summarizes the experimental results of our model versus the baseline models on NYT10d, where bold indicates the best score and underlining indicates the second best score. As can be seen

| Models | NYT10d | | | | |
|---|---|---|---|---|---|
| | AUC | P@100 | P@200 | P@300 | P@M |
| PCNN-ATT | 34.1 | 73.0 | 68.0 | 67.3 | 69.4 |
| RESIDE | 41.5 | 81.8 | 75.4 | 74.3 | 77.2 |
| DISTRE | 42.2 | 68.0 | 67.0 | 65.3 | 66.8 |
| REDSandT | 42.4 | 78.8 | 75.0 | 73.0 | 75.3 |
| CIL | 50.8 | **90.1** | 86.1 | 81.8 | 86.0 |
| HiCLRE | 45.3 | 82.0 | 78.5 | 74.0 | 78.2 |
| HFMRE | **54.0** | 89.0 | **87.0** | **83.3** | **86.4** |

Table 1: Experiment results on distantly supervised datasets.

from the above results: (1) our model HFMRE significantly outperforms all baseline models, improving by 3.2 AUC pts compared to the strongest baseline CIL, even when the CIL uses an additional loss term (MLM loss, CL loss). (2) Our model achieves the best performance on all metrics except for the P @100 metric, which was slightly lower than CIL. (3) Our model achieves a state-of-the-art new result on the NYT10d dataset.

Figure 4 reflects the overall precision-recall profile of our model HFMRE versus the other baseline models on NYT10d. It can be observed from the curves that (1) our HFMRE model has a more convex curve compared to the other baseline models, enclosing almost all other curves, especially in the threshold interval of 0.2 to 0.8, showing higher precision and recall results. (2) Although our curve is slightly lower than the CIL in the initial small interval, which may be the reason why our model is slightly lower than the CIL in the P@100 metric, it is clearly higher for us in the other intervals.

### 4.5 Evaluation on Manually Annotated Set

The test set implemented using the distant supervision hypothesis suffers from data annotation problems and does not accurately reflect model performance, so we further evaluated our method on test sets using manual annotation (NYT10m, Wiki20m). The results of the experiment are shown in Table 2. (1) On NYT10m, our model achieves the best scores on the AUC (63.4) and $\mu$-F1 (62.4) metrics, and the second best score on M-F1(35.7). We can also observe an interesting phenomenon that, in contrast to the NYT10 dataset, the HICLRE model outperforms the CIL model across the board, and we believe that the different levels of semantic interaction and multi-granularity recontextualization of the HICLRE model can play a greater role in annotating accurate data. (2) On Wiki20m, surprisingly

BERT-AVG achieves the best AUC result, which we suspect is caused by the different principles of "N/A" relation labeling in Wiki20m and NYT10 —Wiki20m labels the relations of entity pairs outside of its relation ontology as "N/A", and NYT10 labels entity pairs without relations as "N/A". In other words, our model may predict the entity pair marked as "N/A" in Wiki20m as a relation other than the relation ontology. We achieved the best scores on the M-F1 and $\mu$-F1 metrics, demonstrating that we are still the best performing model in terms of overall accuracy and recall.

In summary, our model has been well generalized and is still able to guarantee performance in real scenarios with manual annotation, which means that our approach can be applied to real problems.

### 4.6 Ablation Study

The following ablation experiments are designed to understand the factors affecting the performance of the HFMRE model: (1) To verify the validity of the circular cosine similarity, instead of using the circular cosine similarity to identify the in-bag noise, we identify the noise by the regular sentence-level attention and then construct a Huffman tree based on the attention score. (2) To verify the validity of the relation query, we use the conventional full connection layer instead of the relation query to obtain the probability score of each relation. (3) To verify the validity of constructing Huffman trees, we perform subsequent relation querying by weighting the sentences in the bag according to the cosine similarity summation, such that HFMRE degenerates into a common multi-instance learning framework. (4) We remove all three components mentioned above so that HFMRE degrades to plain BERT-Att, using BERT-Att as a base criterion for comparison. The results of the experiments on the NYT10m dataset

| Models | NYT10m | | | Wiki20m | | |
|---|---|---|---|---|---|---|
| | AUC | M-F1 | $\mu$-F1 | AUC | M-F1 | $\mu$-F1 |
| BERT-ONE | 58.1 | 33.9 | 61.9 | 88.9 | 81.1 | 81.6 |
| BERT-AVG | 56.7 | 35.7 | 60.4 | **89.9** | 82.0 | 82.7 |
| BERT-ATT | 51.2 | 25.8 | 54.1 | 70.9 | 64.3 | 66.8 |
| CIL* | 59.5 | 36.3 | 60.5 | 88.7 | 81.9 | 82.4 |
| HiCLRE* | 61.4 | **36.9** | 60.9 | 85.6 | 76.7 | 78.0 |
| HFMRE | **63.4** | 35.7 | **62.4** | 89.1 | **82.4** | **83.3** |

Table 2: Experiment results on human-annotated datasets.* means that we have run their model on the datasets which were not covered in the original paper and reported the results. We have obtained the original code of the model from the respective authors and have successfully replicated the results in their paper.

| Models | AUC | Change in AUC |
|---|---|---|
| (1) -CCSC | 60.2 | -3.2 |
| (2) -Relation Query | 57.7 | -5.7 |
| (3) -Huffman Tree | 58.2 | -5.2 |
| (4) BERT-Att | 51.2 | -12.2 |

Table 3: Ablation experiments on the nyt10m dataset.

| Encoder | AUC | P@100 | P@200 |
|---|---|---|---|
| PCNN | 44.6 | 94.0 | 92.5 |
| SPANBERT | 61.0 | 93.0 | 91.5 |
| BERT | **63.4** | **99.0** | **95.0** |

Table 4: Experimental results of different encoders on nyt10m dataset.

are shown in Table 3.

From the experimental results, it is noted that all four models after ablation show a significant performance degradation compared to the complete model HFMRE. From model (1) HFMRE(-CCSC) can confirm our previous speculation that the circular cosine similarity method does focus more on noise than traditional selective attention, and that the intrinsic connections between sentences are of great benefit in identifying noise. Model (2) HFMRE (-Relation Query) performance dropped by 5.7 AUC pts, illustrating the effectiveness of our relation querying step, which indeed can query through the relation matrix to a relation that best fits the entity pair. Model (3) HFMRE(-Huffman Tree) demonstrates that constructing a Huffman tree is an essential process and that the bag-level features constructed by this process can effectively improve the performance of the model. When we ablate the model in its entirety, the performance of the model (4) BERT-Att shows a huge reduction (63.4 → 51.2), which once again illustrates the effectiveness of the modules in our approach. These modules are complementary to each other and are indispensable. We conducted the same experiments on the remaining datasets and the results are presented in Appendix B.

## 4.7 Case Study

### 4.7.1 The Influence of Sentence Encoders

To further understand the effectiveness of our method, we conduct experiments using different encoders to verify that our method is encoder-independent. Taking nyt10m as an illustration, the experimental results are shown in Table 4.

We report experimental results of the model on three different encoders and find interesting phenomena. Firstly, our model performs well on pre-trained models (BERT, SPANBERT), enough to match or even surpass existing methods. Second, on PCNN, the AUC metric of the model gets significantly decreased due to the structural flaws of the PCNN model, however against the common sense the model does not decrease on the P@M (100, 200) metrics, but instead outperforms HFMRE (SPANBERT). Overall, HFMRE achieves good results using different encoding methods, proving that our approach is encoder-independent.

### 4.7.2 The Influence of Different Model Combinations

We have demonstrated the effectiveness of the individual components of the HFMRE model in our ablation study, and in order to gain a deeper understanding of the positive effects of the individual components, we conducted experiments with different combinations of Sentence Encoder = {BERT, SPANBERT, PCNN}, Weight = {Circular Cosine

| Models | AUC |
|---|---|
| (A)PCNN+CCSC+Huffman Tree | 44.6 |
| (B)PCNN+ATT+Weighted Average | 56.8 |
| (C)BERT+ATT+Weighted Average | 51.2 |
| (D)BERT+ATT+ Huffman Tree | 60.2 |
| (E)BERT+CCSC+ Weighted Average | 58.2 |
| (F)BERT+CCSC+Huffman Tree | **63.4** |
| (G)SPANBERT+CCSC+Huffman Tree | 61.0 |

Table 5: Experimental results of different model combinations on the nyt10m dataset.

Similarity (CCSC), Attention Score (ATT)}, and Aggregator = {Weighted Average, Huffman Tree}. Taking nyt10m as an illustration, the experimental results are shown in Table 5.

The analysis reveals that (1) In the case of BERT as an encoder, both our proposed components *CCSC* and *Huffman Tree* can substantially improve the model performance, and using combination (C) as a baseline comparison, it can be observed that the performance of combinations (D), (E), and (F) are all significantly improved. (2) Our model architecture is suitable for BERT or its variant models (SPANBERT, etc.), but not suitable for PCNN. We suspect that this is due to the convolutional kernel, which messes up the overall semantics of a sentence. The features encoded by PCNN are composed of many local semantics, rather than BERT encoding features from the perspective of the global semantics of the whole sentence. Because of this coding characteristic of PCNN, the original position of the sentence in the vector space changes, so that the circular cosine similarity can not distinguish the noise effectively.

## 5    Conclusion and Outlook

In this work, we propose a relation extraction model HFMRE for distantly supervised tasks to address the shortcomings of existing aggregation strategies. Moving away from the traditional mindset, HFMRE innovatively treats sentences as leaf nodes and continuously identifies noise and aggregates positive example features within the bag as the sentences are used to construct Huffman trees. Experiments have demonstrated the effectiveness of our method, as compared to existing baseline models, our HFMRE can significantly identify in-bag noise and is able to find an excellent instance that represents a bag-level feature.

In future work, we see more possibilities for the

HFMRE architecture. For example, HFMRE can construct different Huffman trees for different relations, i.e. 53(Number of relations in NYT10d dataset) Huffman trees can be constructed within a bag, and then relation querying can be performed on the 53 outstanding instances. Alternatively, positive pair Huffman trees and negative pair Huffman trees can be constructed separately to further improve model performance through contrastive learning.

## Limitations

Although our HFMRE is able to find excellent bag-level features, there are still some limitations. (1) The complexity of our method is determined by the number of nodes needed to construct a Huffman tree, so when there are too many sentences in a multi-instance bag, the complexity of the model skyrockets, which means that more GPU memory is needed. In practice, therefore, we limit the number of nodes and we select the required Huffman nodes from a multi-instance bag by random sampling. Through experimentation, we recommend that selecting 10 and less number of sentences from a bag as nodes is the optimal choice.(2) Our HFMRE has only been experimented on monolingual datasets, and its performance on multilingual datasets has yet to be verified.

## Acknowledgements

This work was supported by the National Key R&D Plan of China (2022YFF0608002-01).

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

## A  Hyper-parameter Setting and Implementation Details

| Hyper-parameters | Value |
|---|---|
| hidden size | 786 |
| max_seq_len | 258 |
| epochs | 3/3/5 |
| learning rate | 2e-5 |
| weight_decay | 1e-5 |
| train batchsize | 16 |
| test batchsize | 16 |
| Activation function | tanh |
| optimizer | adamw |
| num_nodes | 1/6/6 |

Table 6: Detailed settings for hyper-parameters. num_nodes represents the maximum number of nodes needed to construct a Huffman tree.

**Implementation details**: The IDE used for the experiments in this paper is Pycharm2021 Professional Edition. the PyTorch version is 1.9.1; CUDA version is 11.6; CUDNN version 10.2.The model training and inference were performed on an NVIDIA A100-SMX with 40GB of GPU memory and 16GB of CPU memory.

HFMRE takes approximately 40 minutes to train each epoch for the NYT10d dataset, 30 minutes for the NYT10m dataset, and 50 minutes for the Wiki20m dataset.

For parameter tuning, we take a manual tuning approach, with tuning intervals of $\{8, 16, 32\}$ for batch size, $\{1, 2, \cdots, 15\}$ for num_nodes, and $\{1e-5, 2e-5\}$ for learning rate.

## B  Ablation Study

| Models(AUC) | NYT10d | NYT10m | Wiki20m |
|---|---|---|---|
| (1) -CCSC | 52.9(-1.1) | 60.2(-3.2) | 82.8(-6.3) |
| (2) -Relation Query | 50.6(-3.4) | 57.7(-5.7) | 80.8(-8.3) |
| (3) -Huffman Tree | 52.8(-1.2) | 58.2(-5.2) | 86.7(-2.8) |
| (4) BERT-Att | 27.8(-26.2) | 51.2(-12.2) | 70.9(-18.2) |

Table 7: Performance of different ablation models on AUC metrics.

## C Datasets Statistics

| Datasets | Train | | | Validation | | | Test | | | #Rel |
|---|---|---|---|---|---|---|---|---|---|---|
| | #facts | #sents | N/A | #facts | #sents | N/A | #facts | #sents | N/A | |
| NYT10d | 18,409 | 522,611 | 74% | - | - | - | 1,950 | 172,448 | 96% | 53 |
| NYT10m | 17,137 | 417,893 | 80% | 4,062 | 46,422 | 80% | 3,899 | 9,744 | 32% | 25 |
| Wiki20m | 157,740 | 698,721 | 59% | 17,485 | 64,607 | 73% | 56,000 | 137,986 | 25% | 81 |

Table 8: Statistics of three datasets.

## D Huffman tree Algorithm

---
**Algorithm 1** Construct Huffman tree inside the bag

---
**Require:** HuffmanNode $\{name, weight, value\}$ ▷ *Creating leaf nodes,each leaf node corresponds to a name, weight and value.*
**Ensure:** The Huffman Tree
 1: createLeafNodes(nodes, name, weight, value); ▷ *nodes, i.e. sentences within a bag.*
 2: PriorityQueue queue; ▷ *Creating a Huffman tree node queue.*
 3: **for** each node in nodes **do**:
 4:    enqueue(queue, node);
 5: **end for**
 6:                  ▷ *Constructing Huffman trees.*
 7: **while** size(queue) > 1 **do**:
 8:    ▷ *Remove the two nodes with the lowest weight from the queue.*
 9:    node1 = dequeue(queue);
10:    node2 = dequeue(queue);
11:    ▷ *Create a new node as their parents, default right child is the smaller of weight.*
12:    parent = createNode(node1 , node2);
13:    parent.left = node1;
14:    parent.right = node2;
15:             ▷ *Putting parent nodes in the queue.*
16:    enqueue(queue, parent);
17: **end while**
18: **return** dequeue(queue); ▷ *The last node left in the queue is the root node.*
19:  **void** createNode(left,right):
      node.name = Max(queue[i].name)+1;
      node.weight = CCSC(queue);
      node.feature = left.feature⊕right.feature;

---