# OpenReview forum: "HFMRE: Constructing Huffman Tree in Bags to Find Excellent Instances for Distantly Supervised Relation Extraction"
_EMNLP/2023/Conference — EMNLP 2023 Findings_

### Official Review · Reviewer_kjoj · 2023-08-01

**Soundness:** 3

**Excitement:**

3: Ambivalent: It has merits (e.g., it reports state-of-the-art results, the idea is nice), but there are key weaknesses (e.g., it describes incremental work), and it can significantly benefit from another round of revision. However, I won't object to accepting it if my co-reviewers champion it.

**Paper Topic And Main Contributions:**

This paper focuses on the Distant Supervised Relation Extraction (DSRE) task and propose a Huffman tree-based Multi-Instance Learning (MIL) method. Experiments show the remarkable effectiveness of the proposed method on several popular DSRE datasets.

**Questions For The Authors:**

Q1: The experiments in the paper are all conducted with BERT sentence encoder. However, the proposed method is encoder-independent. So, I’m wondering how does the HFMRE model perform if the sentence encoder is switched to PCNN. This result may help proving your hypothesis mentioned in Line 107.

Q2: The two contributions of this paper focus on weight computation and bag feature aggregation in MIL. I’m hoping to see more experimental results on more complete model combinations. For example, sentence encoder={BERT, PCNN}, weight={cosine, circular cosine, attention score}, aggregator={weighted average, Huffman tree}. These results may lead to more solid contributions.

Q3: In abstract you claimed that existing strategies rely on third-party vectors to determine what is noise, but when merging tree nodes HFMRE still uses a thirty-party matrix W_s, which decides how much of each input component (either noise or not) contributes to the output representation. Is this noise-deciding method still relying on third-party parameters? Would you please explain the key difference between HFMRE and existing methods from this perspective?


**Reasons To Accept:**

This paper proposes the circular cosine similarity to measure the inherent associations between sentences and the Huffman tree-based method to capture bag representations. The proposed model is novel and outperforms baseline models on several benchmark datasets.

**Reasons To Reject:**

Some of the important experiments are missing, such as comparison of different sentence encoder, weight computing method, and bag feature aggregator. See details in Q1 and Q2 of “Questions For The Authors”.

**Reproducibility:**

4: Could mostly reproduce the results, but there may be some variation because of sample variance or minor variations in their interpretation of the protocol or method.

**Reviewer Confidence:**

3: Pretty sure, but there's a chance I missed something. Although I have a good feel for this area in general, I did not carefully check the paper's details, e.g., the math, experimental design, or novelty.

**Typos Grammar Style And Presentation Improvements:**

Line 502: Figure 5 -> Figure 4

Line 521: Table 5 -> Table 2

---

> ### Author Rebuttal · Authors · 2023-08-28
>
> We greatly appreciate your careful review of our paper, and any deficiencies, comments, or confusion you point out will be answered below.
>
> $\textbf{1.	Response on Q1}$
>
> |  Encoder | AUC | P@100 |  P@200 | P@300 |
> | :---- | ----: | :----: | :---- | ----: |
> | HFMRE(PCNN) | 44.6| 94.0| 92.5| 91.0|
> |HFMRE(SPANBERT)| 61.0 | 93.0 | 91.5 | 88.7 |
> | HFMRE(BERT) | 63.4| 99.0 | 95.0| 93.7  |
>
> We report experimental results of the model on three different encoders and discover some interesting phenomena. Firstly, our model performs well on pre-trained models (BERT, SPANBERT), enough to match or even surpass existing methods. Second, on PCNN, the AUC metric of the model gets significantly decreased due to the structural flaws of the PCNN model, however, against common sense the model does not decrease on the P@M (100, 200, 300) metric, but instead outperforms HFMRE (SPANBERT).
>
> $\textbf{2.	Response on Q2}$
>
> |  Combination | AUC |
> | :---- | ----: |
> | PCNN+CCSC+Huffman tree | 44.6|
> |PCNN+ATT+weighted average| 56.8 |
> | BERT+ATT+weighted average | 51.2|
> | BERT+ATT+ Huffman tree | 60.2|
> | BERT+CCSC+ weighted average | 58.2|
> | BERT+CCSC+Huffman tree | **63.4**|
> | SPANBERT+CCSC+Huffman tree | 61.0|
>
> We conducted experiments with different combinations of Sentence Encoder = {BERT, SPANBERT, PCNN}, Weight = {Circular Cosine（CCSC）, Attention Score (ATT)}, and Aggregator = {Weighted Average, Hoffman Tree}, and the results of the experiments are shown in the table above.
>
> Through analysis, it is found that our model architecture is suitable for BERT or its variant models (SPANBERT, etc.), but not suitable for PCNN. We suspect that this is due to the convolutional kernel, which messes up the overall semantics of a sentence. The features encoded by PCNN are composed of many local semantics, rather than BERT encoding features from the perspective of the global semantics of the whole sentence. Because of this coding characteristic of PCNN, the original position of the sentence in the vector space changes, so that the Circular cosine similarity can not distinguish the noise effectively.
>
> $\textbf{3.	Response on Q3}$
>
> We consider that existing methods rely on third-party vectors to determine what is noise, so the HFMRE model uses circular cosine similarity to determine noise, and the lower the circular cosine similarity, the more likely it is to be recognized as noise (this is explained in the manuscript at lines 352-364). The circular cosine similarity does not train any parameters and does not depend on third-party vectors.
>
> For merging tree nodes we use a parameter matrix WS, which serves only to reduce the dimensionality of the vectors (1536 to 768). In this paper the hidden dimension of the nodes is 768, so we are going to use the matrix WS to reduce the dimension of the newly merged nodes to 768.
>
> For determining the contribution of each input component to the output representation, we are using a hyper-parameter α to control the effect of noise.

---

### Official Review · Reviewer_LrYw · 2023-08-02

**Typos Grammar Style And Presentation Improvements:** In L170, 'it' should be changed for '…
**Soundness:** 2

**Excitement:**

2: Mediocre: This paper makes marginal contributions (vs non-contemporaneous work), so I would rather not see it in the conference.

**Paper Topic And Main Contributions:**

The authors propose a new architecture, HFMRE, for Distantly Supervised Relation Extraction. The authors focus on how to select the important information from the bags. Firstly, HFMRE uses the idea of the Harman tree algorithm, which treats sentences as nodes to be selected, discovers noise and aggregates sentence information. Then, using the concept of circular cosine similarity to quickly and effectively discriminate potentially noisy data in the bag. The experimental result shows that proposed method is effective.

**Questions For The Authors:**

1 The author has been emphasizing the inherent association, what is the inherent association?
2 Besides the validity of the method in the overall experimental results, are there more detailed indicators or examples to demonstrate the validity of the proposed method?
3 Why is there no comparison with the large language model?

**Reasons To Accept:**

Experimental results are solid.

**Reasons To Reject:**

Poor written, methods are not novel, and motivation is not very novel.

**Reproducibility:**

2: Would be hard pressed to reproduce the results. The contribution depends on data that are simply not available outside the author's institution or consortium; not enough details are provided.

**Reviewer Confidence:**

3: Pretty sure, but there's a chance I missed something. Although I have a good feel for this area in general, I did not carefully check the paper's details, e.g., the math, experimental design, or novelty.

---

> ### Author Rebuttal · Authors · 2023-08-28
>
> We greatly appreciate your careful review of our paper, and any deficiencies, comments, or confusion you point out will be answered below.
>
> $\textbf{1.	Response on the inherent association mentioned in the paper}$
>
> Intrinsic association actually refers to the distribution of the positions of the sentences within the bag in the vector space. Simply put, when two sentences are more semantically similar, their positions in the vector space are closer. Conversely, the more distant the positions in the vector space. We summarise the proximity of sentences in vector space as the strength of the intrinsic connection between sentences.
> In our paper, we further upgraded the notion of intrinsic association to the notion of circular cosine similarity by taking the sum of the distances of a sentence from all other sentences in the bag as its own circular cosine similarity. We explain its conceptual strengths and why it works in lines 352-364 of the paper.
>
> $\textbf{2.	Response on the use of other indicators or experiments to demonstrate the validity of the methodology}$
>
> (1) We conducted ablation experiments in our paper to demonstrate the effectiveness of the individual modules in the methodology.
>
> (2) We performed additional experiments to compare the results of experiments using different encoders. The nyt10m dataset is used as an example.
>
> |  Encoder | AUC | P@100 |  P@200 | P@300 |
> | :---- | ----: | :----: | :---- | ----: |
> | HFMRE(PCNN) | 44.6| 94.0| 92.5| 91.0|
> |HFMRE(SPANBERT)| 61.0 | 93.0 | 91.5 | 88.7 |
> | HFMRE(BERT) | 63.4| 99.0 | 95.0| 93.7  |
>
> We report experimental results of the model on three different encoders and discover some interesting phenomena. Firstly, our model performs well on pre-trained models (BERT, SPANBERT), enough to match or even surpass existing methods. Second, on PCNN, the AUC metric of the model gets significantly decreased due to the structural flaws of the PCNN model, however, against common sense the model does not decrease on the P@M (100, 200, 300) metric, but instead outperforms HFMRE (SPANBERT).
>
> (3) We additionally conducted experiments (using nyt10m as an example) to form a case study with the aim of comparing the quality of instance selection for different methods. The experiment is divided into two parts: one for the quality of the selected sentence-level features and one for the quality of the selected bag-level features.
>
> |  Sentence-level | AUC | μ-F1 |  Bag-level | AUC | μ-F1 |
> | :---- | ----: | :----: | :---- | ----: | :----: |
> | BERT-One | 58.1  | 61.9 | BERT-Att | 51.2 | 54.1 |
> |OUR | **60.9** | **62.1** | BERT-Avg | 56.7 | 60.4|
> | - | - | - | OUR | **63.4**  | **62.4** |
>
> According to the experimental results, our method outperforms previous methods both in terms of the quality of instances selected for individual sentences and the quality of bag-level features selected.
>
> (4) We conducted additional experiments to compare the impact of different model combinations on the experimental results (using the nyt10m dataset as an example).
>
> |  Encoder | AUC |
> | :---- | ----: |
> |  PCNN+CCSC+Huffman tree| 44.6 |
> |  PCNN+ATT+weighted average| 56.8|
> |  BERT+ATT+weighted average| 51.2|
> |  BERT+ATT+ Huffman tree| 60.2|
> |  BERT+CCSC+ weighted average| 58.2|
> |  BERT+CCSC+Huffman tree| **63.4**|
> |  SPANBERT+CCSC+Huffman tree| 61.0 |
>
> We conducted experiments with different combinations of Sentence Encoder = {BERT, SPANBERT, PCNN}, Weight = {Circular Cosine（CCSC）, Attention Score (ATT)}, and Aggregator = {Weighted Average, Hoffman Tree}, and the results of the experiments are shown in the table above.
> Through analysis, it is found that our model architecture is suitable for BERT or its variant models (SPANBERT, etc.), but not suitable for PCNN. We suspect that this is due to the convolutional kernel, which messes up the overall semantics of a sentence. The features encoded by PCNN are composed of many local semantics, rather than BERT encoding features from the perspective of the global semantics of the whole sentence. Because of this coding characteristic of PCNN, the original position of the sentence in the vector space changes, so that the Circular cosine similarity can not distinguish the noise effectively.
>
> $\textbf{3.	Response on why comparisons with larger models were not made}$
>
> Large models are a hot topic in current research, and we have two reasons for not comparing with large models:
>
> (1) Firstly, we think it is unfair to compare with large models, the methods we compare in our paper are based on models such as BERT-base, CNN, etc., which have much smaller number of parameters than large models.
>
> (2) Secondly, to the best of our knowledge, we are not aware of any large model studies on distantly supervised relation extraction. Therefore we have no baseline model to compare.
>
> $\textbf{4.	Response on Soundness}$
>
> In the Soundness metrics, it was mentioned that some of the claims in the paper not being fully supported, and since you didn't specify what experiments were missing, we added three additional experiments (shown in the response above,), and combined them with the experiments in the paper, we hope that these experiments will meet your expectations.
>
> There is also a huge problem mentioned in the Soundness about the technique and methodology in the paper. We describe our methodology in detail in the paper with reasonable explanations, and in terms of technical implementation, we provide the source code in the supporting material.
>
>
> $\textbf{5.	Response on Reproducibility}$
>
> We state that all data used in the paper are publicly available.
>
> In the Reproducibility indicator, you pointed out that the experimental results are difficult to reproduce, that the experimental data are not available at institutions or consortia other than the authors, and that insufficient detail was provided.
>
> This view may be due to the fact that you have ignored the appendices and supporting materials of our paper. In the appendix of the paper, the parameter settings as well as the implementation details are explained in detail, and the download links to the experimental data and the environmental requirements are provided in the source code.
>
>
> $\textbf{6.	Response on the novelty of the methodology and motivation in the paper}$
>
> The first is the novelty of the motivation --- in the field of distantly supervised relation extraction, a large amount of mislabelled data is unavoidable, as datasets are labeled distantly against a corpus and a knowledge base. As a result, getting models to learn to recognize noise has been an unavoidable problem since the direction was proposed to the present day, and even beyond. In other words, getting the model to identify noise and removing it is the fundamental task to be addressed by distantly supervised relational extraction. Existing state-of-the-art methods, whether using reinforcement learning, comparative learning, or various multi-level, multi-granularity coding modules, ultimately aim at reducing the noise problem. Therefore, we do not feel that the claim of novelty of motivation is valid.
> The next is the novelty of the method --- the innovation of our method lies in two points, one is that we abandon the use of third-party vectors to identify in-bag noise, and instead use the intrinsic associations, so that we do not need to train additional model parameters, and to the best of our knowledge we are the first to not use the trained vectors/matrices to guide the model in identifying the noise. Second, our method is a continuous iterative process in identifying noise and aggregating bag features, and we re-identify the data that is most likely to be noisy with every node we merge. Existing methods all perform noise identification only once and then directly aggregate bag features based on the weights assigned.

---

### Official Review · Reviewer_sGH7 · 2023-08-04

**Typos Grammar Style And Presentation Improvements:** 1)  Section 3.4 The process is denote…
**Soundness:** 3

**Excitement:**

3: Ambivalent: It has merits (e.g., it reports state-of-the-art results, the idea is nice), but there are key weaknesses (e.g., it describes incremental work), and it can significantly benefit from another round of revision. However, I won't object to accepting it if my co-reviewers champion it.

**Paper Topic And Main Contributions:**

This paper presents a methodology for selecting instances for training models in relation extraction tasks.  It assumes a multi-instance learning setup and proposed weighting strategies of samples in a bag. The proposed method shows improvement over other state-of-the-art methodologies.

**Questions For The Authors:**

1) how would the conclusion look if a different base embedding (other than BERT) is used?

2) In line 502, there was some discussion around Area under PRC. can the paper shed some light on the AUPRC numbers of this method?

**Reasons To Accept:**

1) constructing the Huffman tree using circular cosine similarity seems novel; it also shows potential in the ablation study

2) provides state-of-the-art results in most of the datasets.

**Reasons To Reject:**

1) No time complexity results are presented for the paper. A thorough analysis of those aspects will help the community.

2) As this paper is about selecting good instances from a bag. But, there were no case study presented to compare side by side different methods on their quality of selections.



**Reproducibility:**

4: Could mostly reproduce the results, but there may be some variation because of sample variance or minor variations in their interpretation of the protocol or method.

**Reviewer Confidence:**

4: Quite sure. I tried to check the important points carefully. It's unlikely, though conceivable, that I missed something that should affect my ratings.

---

> ### Author Rebuttal · Authors · 2023-08-28
>
> We greatly appreciate your careful review of our paper, and any deficiencies, comments, or confusion you point out will be answered below.
>
> $\textbf{ 1.	Time complexity analysis }$
>
> Assuming that there are n sentences in the bag, the complexity of merging leaf nodes is o(1), then constructing a Huffman tree has to be looped n times, and the time complexity of each loop is no more than log2n, so the time complexity of constructing a Huffman tree is $nlog_{2}^{n}$.
>
> Because the circular cosine similarity has to be calculated once after each node is merged, the first time there are n nodes, each node has to be calculated once, so the first time is n, the second time there are n-1 nodes, so it is n-1, since the total number of loops is n times, the time complexity of the circular cosine similarity is n+(n-1)+(n-2)+... ...+1=n(n+1)/2.
>
> So the time complexity of HFMRE is $nlog_{2}^{n}+ n(n+1)/2$.
>
> $\textbf{ 2.	Response to case studies on different approaches to selecting excellent examples }$
>
> We additionally conducted experiments (using nyt10m as an example) to form a case study with the aim of comparing the quality of instance selection for different methods. The experiment is divided into two parts: one for the quality of the selected sentence-level features and one for the quality of the selected bag-level features.
>
> |  Sentence-level | AUC | μ-F1 |  Bag-level | AUC | μ-F1 |
> | :---- | ----: | :----: | :---- | ----: | :----: |
> | BERT-One | 58.1  | 61.9 | BERT-Att | 51.2 | 54.1 |
> |OUR | **60.9** | **62.1** | BERT-Avg | 56.7 | 60.4|
> | - | - | - | OUR | **63.4**  | **62.4** |
>
> According to the experimental results, our method outperforms previous methods both in terms of the quality of instances selected for individual sentences and the quality of bag-level features selected.
>
> $\textbf{3.	Response on using different encoders (except BERT) in Q1}$
>
> |  Encoder | AUC | P@100 |  P@200 | P@300 |
> | :---- | ----: | :----: | :---- | ----: |
> | HFMRE(PCNN) | 44.6| 94.0| 92.5| 91.0|
> |HFMRE(SPANBERT)| 61.0 | 93.0 | 91.5 | 88.7 |
> | HFMRE(BERT) | 63.4| 99.0 | 95.0| 93.7  |
>
> We report experimental results of the model on three different encoders and discover some interesting phenomena. Firstly, our model performs well on pre-trained models (BERT, SPANBERT), enough to match or even surpass existing methods. Second, on PCNN, the AUC metric of the model gets significantly decreased due to the structural flaws of the PCNN model, however, against common sense the model does not decrease on the P@M (100, 200, 300) metric, but instead outperforms HFMRE (SPANBERT).
>
> $\textbf{4.	Response to the question about the PR curve graph at line 502 of the paper, mentioned in Q2}$
>
> This paper's discussion of the PR curve at line 502 actually revolves around Figure 4, not Figure 5. We apologize that our error caused you to be confused when reading the manuscript.

---

### Meta-Review · Area_Chair_XEWp · 2023-09-05

**Recommendation:** 3
**Best Paper Recommendation:** No

**Metareview:**

The paper proposes a method for performing distantly supervised relation extraction based on Huffman tree multi-instance learning and circular cosine similarity, which considers intrinsic associations between sentences using BERT as an encoding, and demonstrates good experimental results. The reviewers agree that the proposed method is novel and the presented results are solid. They are mainly concerned with a need for further analysis and experimentation in order to strengthen the paper. Some further results and details are presented by the authors in their response, including the use of different sentence encoders (SPANBERT and PCNN), a description of the time complexity of their method, and another baseline comparison. The paper appears to be mostly well executed as the results are robust, which the provided new experiments will supplement.

**Meta-Review:**

The paper proposes a method for performing distantly supervised relation extraction based on Huffman tree multi-instance learning, and demonstrates good experimental results. The reviewers agree that the proposed method is novel and the presented results are solid. They are mainly concerned with a need for further analysis and experimentation in order to strengthen the paper. Some further results are presented by the authors in their response, but reviewers still feel that further revision is necessary.

---

### Decision · Program_Chairs · 2023-10-07

**Decision:**

Accept-Findings

**Comment:**

The paper proposes a method for performing distantly supervised relation extraction based on Huffman tree multi-instance learning and circular cosine similarity, which considers intrinsic associations between sentences using BERT as an encoding, and demonstrates good experimental results. The reviewers agree that the proposed method is novel and the presented results are solid. They are mainly concerned with a need for further analysis and experimentation in order to strengthen the paper. Some further results and details are presented by the authors in their response, including the use of different sentence encoders (SPANBERT and PCNN), a description of the time complexity of their method, and another baseline comparison. The paper appears to be mostly well executed as the results are robust, which the provided new experiments will supplement.